# Research on Evaluation Method of Electric Vehicle Wireless Charging Interoperability Based on Two Parameter Representation

**Lin Sha \*, Jiangang Liu**  **and Zhixin Chen**

Tianjin Key Laboratory of Intelligent Control of Electrical Equipment, Tiangong University, Xiqing District, Tianjin 300387, China
* Correspondence: shalin@tiangong.edu.cn

**Abstract:** The interoperability of wireless charging for electric vehicles refers to the radio energy transmission that meets the performance and function requirements of different manufacturers and different models of electric vehicles on the premise of meeting the relevant requirements. If it fails to meet the requirements, the wireless charging system of electric vehicles has difficulty to realize interconnection and low charging efficiency, Therefore, how to evaluate the interoperability is a key issue in the promotion of electric vehicle wireless charging. In this paper, an interoperability evaluation method based on two parameters is proposed. The interoperability impedance plane is constructed by the system detuning coefficient A. The comprehensive evaluation of different compensation networks and coupling coils is realized; the power characteristic impedance $\varepsilon$ is obtained by analyzing and calculating the relationship between the transmission power of the system while the system impedance, and the transmission power evaluation of the wireless power transmission system is realized. At the same time, according to simulation and experiment, it was verified that A meets the interoperability requirements when A is in the range of $(-0.62, 0.62)$ in the aligned position and $(-0.75, 0.75)$ in the offset position. When the input voltage is 200 V, when $\varepsilon$ satisfies $0.1925 \geq \varepsilon > 0.0925$, the system WPT2 power level transmission interoperability requirements are met. The method in this paper can guide the interoperability evaluation of electric vehicle wireless charging.

**Keywords:** wireless charging; interoperability; dual parameter characterization; compensation network; coupling coil

## 1. Introduction

As a new type of environmentally friendly vehicle powered by batteries, electric vehicles play an important role in reducing operating costs from fossil energy consumption and environmental pollution. There are two main charging methods for electric vehicles on the market: wired charging and wireless power transfer (WPT) [1–3]. Wired power transmission is relatively common and has the advantages of low loss and strong anti-interference ability. However, because the wired charging device must be in contact with the device to be charged, there needs to be structures such as plugs and sockets. Therefore, it is inconvenient to use, prone to sparks, leakage, easily wears, and electric shock may generate heat, fire, with power limitations. As a new charging method for electric vehicles, WPT technology does not require contact between the charger and the device to be charged, so there is no physical charging interface wear, sparks, leakage, and other problems, but which can meet the charging needs of electric vehicles under different conditions. At present, countries around the world are scrambling to develop WPT technology, which has become a research hotspot in academia and industry [4,5].

With the promotion of wireless power transmission technology for electric vehicles, there are more and more system suppliers, more and more models covered, and more

diverse usage scenarios. Therefore, solving the interoperability between different brands of products has become the key to the difficult problems of standard development [6–9]. The WPT interoperability of electric vehicles means that the ground equipment and in-vehicle equipment produced by different WPT equipment manufacturers of electric vehicles can meet the performance requirements under the premise of ensuring charging safety, the specified power level, the ground clearance type, and the functional requirements for wireless power transfer [10–12].

Regarding interoperability, a lot of research work has been conducted by enterprises and scholars around the world. Scholars at the University of Michigan have studied the interoperability between unipolar coils and double-D coils, and they have concluded that the coupling coefficient between the two is related to offset, primary and secondary shape, and size: WiTricity introduced high-power EV high-performance wireless charging systems are designed to provide maximum interoperability for a wide range of car brands and vehicle types; D. Thrimawithana of the University of Auckland in New Zealand has proposed a hybrid system with increased coil offset that uses the LCL and CL resonant compensation network, effectively overcoming the adverse effects caused by misalignment between coils, and is suitable for static and dynamic wireless charging; the Society of Automotive Engineers (SAE) is the only international organization currently performing WPT system testing for electric vehicles. In November 2017, the "SAE J2954$^{TM}$ Lightweight PHEV/EV Wireless Power Transmission and Positioning Method Standard Recommended Practice" was released [13]: in July 2016, WiTricity's new high-power EV high-performance WPT system aimed to provide maximum interoperability, in order to be suitable for a variety of car brands and vehicle types, from sports cars and sedans to off-road vehicles, including all-electric vehicles and plug-in hybrid vehicle platforms; the WPT system interoperability seminar for electric vehicles held in Beijing on 5 September 2017, with the participation of relevant experts from China and Germany, built an interoperability test platform and conducted impedance matching tests on ground transmitters and on-board receivers to determine the scope of interoperability. In addition to this, there is a lot of research and standards development driving the development of wireless charging interoperability [14–16].

The interoperability of the electric vehicle WPT system is be divided into compensation network interoperability and coil interoperability in the electrical field, as shown in Figure 1. The relative position change of coils and the difference of compensation network parameters will affect the interoperability of the system. Therefore, the evaluation of interoperability mainly starts from these two aspects. The current mainstream evaluation methods include the following three types:

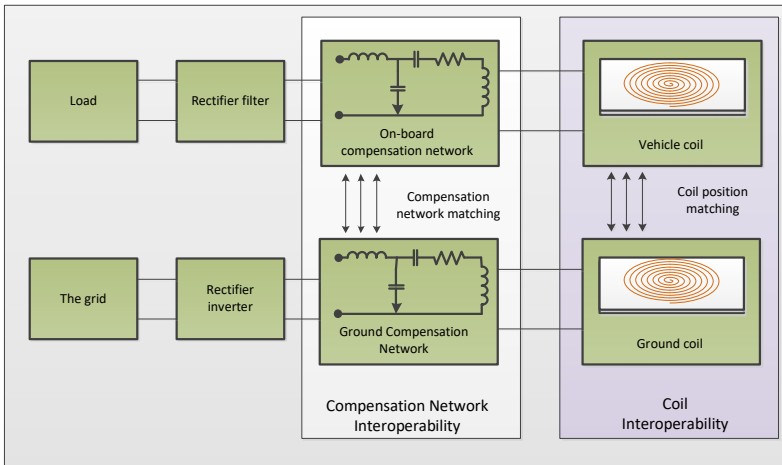

**Figure 1.** Schematic diagram of WPT interoperability system for electric vehicles.

1.  "Output power-transmission efficiency" evaluation method

This method is the most basic and essential evaluation method of system interoperability. By judging the output power $p_{out}$ and transmission efficiency $\eta$ of the electric vehicle WPT system when $p_{out}$ and $\eta$ meet the prescribed power level and efficiency requirements, the system has interoperability, otherwise it does not have interoperability. Although the effect of the evaluation method is visual, it needs to test different working states and test points of the system, which leads to a huge amount of testing work. At present, the most mainstream and complete WPT standard for electric vehicles is SAE J2954 of the American Society of Automotive Engineers. The $p_{out}$ and η interoperability are defined in Table 1, the classification of the system power level is shown in Table 2.

**Table 1.** Classification of WPT system for electric vehicles in SAE J2954.

| Standard | Power Level Pin (kw) | Value Range |
|:---:|:---:|:---:|
| SAE J2954 | WPT1 | $p \leq 3.7$ |
| | WPT2 | $3.7 \leq p \leq 7.7$ |
| | WPT3 | $7.7 \leq p \leq 11.1$ |
| | WPT4 | $11.1 \leq p \leq 22$ |

**Table 2.** Provisions on $P_{out}$ and $\eta$ in SAE J2954.

| Standard | | | Output Power $P_{out}$ (kW) | | | | Transmission Efficiency $\eta$ (%) |
|:---:|:---:|:---:|:---:|:---:|:---:|:---:|:---:|
| | Ground end | | Ground equipment | | | | |
| | Vehicle terminal | | WPT1 | WPT2 | WPT3 | WPT4 | |
| SAE J2954 | On board equip-ment | WPT1 | support | support | support | undetermined | Face to face: $\eta$ More than 85% offset: $\eta \geq 80\%$ |
| | | WPT2 | support | support | support | undetermined | |
| | | WPT3 | support | support | support | undetermined | |
| | | WPT4 | undetermined | undetermined | undetermined | undetermined | |

2.  "Coupling coefficient-quality factor" evaluation method

The coupling coefficient K and quality factor Q can directly reflect the quality characteristics of the mutual inductance coil and compensation network [17]. The disadvantage is that with the increase of the number of coils and the compensation network order and the deepening of the complexity of coil design, the calculation and measurement difficulty of K and Q values will gradually increase, which will lead to the decline of the accuracy of the obtained data and affect the evaluation results of the system.

3.  "Characteristic impedance" evaluation method

By selecting the impedance value of a device or two ports of the system, the characteristics of the system are characterized to achieve the purpose of reflecting the system characteristics [18–20]. The advantage of this method is that researchers can select the corresponding characteristic impedance according to different research needs, which can clearly express the researcher's purpose, and the calculation and measurement are relatively convenient, flexible, and easy to operate. The disadvantage of the WPT system is that there are many complex relationships such as magnetic field coupling, electric field coupling, and field circuit coupling. It is difficult to fully characterize the interoperability between compensation networks and coils simply by measuring a certain electrical parameter or by inaccurate selection of the characterization parameters and the coil, which is especially true for the high-order compensation circuit. So how to accurately select the characteristic impedance as the characterization parameter of the system properties becomes the key to the popularization of this method.

## 2. System Interoperability Modeling Analysis

Combined with the comparison and analysis of the current evaluation methods of electric vehicle WPT interoperability, the "characteristic impedance" evaluation method was selected as the evaluation method in this paper. Combined with the characteristics of this method, the interoperability of the electric vehicle WPT system was modeled.

### 2.1. WPT System Compensation Network Selection

The LCC-LCC compensation structure has the advantages of strong anti-drift ability, flexible parameter design, and high system efficiency; At the same time, this compensation structure is the only compensation structure recommended by GB/T 38775, and it is also the mainstream compensation structure selected in SAE J2954. Its unique advantages and attention at home and abroad have made the LCC-LCC compensation structure become the main compensation structure for the WPT system of electric vehicles for the future. Figure 2 shows the structure of the WPT compensation network for electric vehicles in SAE J2954. Therefore, this paper targets the interoperability of the WPT system of electric vehicle under the LCC-LCC compensation network.

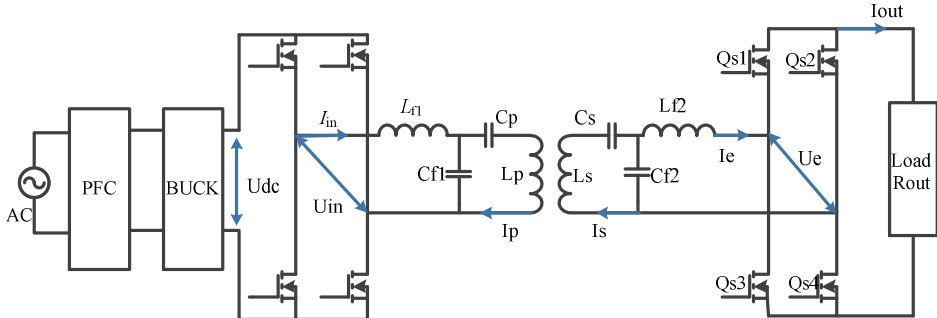

**Figure 2.** WPT compensation network for electric vehicles in SAE J2954.

### 2.2. WPT Interoperability Modeling Analysis

In the process of electric vehicle parking in the charging area, due to the different manufacturers and product models of ground-end charging equipment and vehicle-end equipment, there will be differences between the electrical parameters of the system, which will affect the circuit coupling relationship at both ends, and then affect the interoperability of the system; On the other hand, due to human factors or the limitations of current automatic parking technology, it is difficult to realize direct parking between the ground end coil and the vehicle end coil. The offset between the coils will lead to the change of coil self inductance and mutual inductance, which will affect the transmission efficiency of the system and also affect system interoperability. In view of the influence of coil offset and compensation network parameters, at the same time, considering the inconvenience of the high-order compensation network in parameter calculation and formula derivation, this paper uses VA and GA's vehicle terminal impedance $Z_{VA}$, ground terminal impedance $Z_{GA}$, and the system terminal impedance $Z_{SA}$, including the VA and GA parts which are used to represent the system variation difference. $Z_{VA}$ is defined as the impedance of two ports between the two ends of the vehicle end coil and the load. The secondary side compensation parameters ($C_2$, $C_{f2}$, $L_{f2}$) can be normalized; $Z_{GA}$ is the two port impedance between the two ends of the coil on the ground and the load, which can be used to characterize the coupling of the coil ($k$, $L_1$, $L_2$) and the resonance of the primary resonators ($C_1$, $C_{f1}$, $L_{f1}$); $Z_{SA}$ is a two-port impedance that includes $Z_{VA}$, $Z_{GA}$, and ground-side compensation at both ends of the voltage source, which can characterize the overall resonance of the vehicle-mounted terminal and the ground-side terminal.

This paper mainly studies the effect of circuit detuning on reactive power loss caused by different compensation network parameters and vehicle offset. Therefore, in order to highlight the key points, the active power loss generated by line impedance is ignored. Based on this, the LCC-LCC compensation network is constructed, as shown in Figure 3.

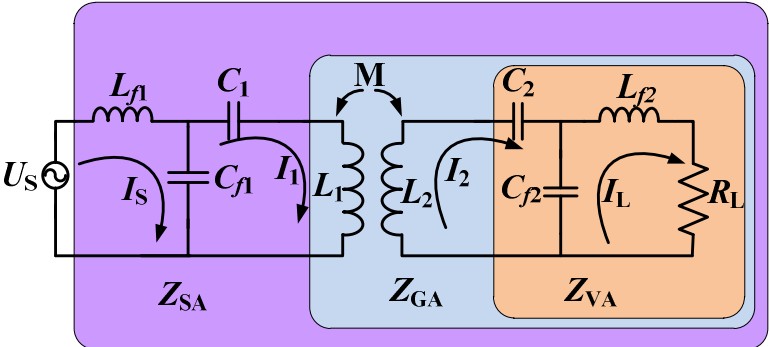

**Figure 3.** LCC-LCC compensation network topology.

At this point, the expression of vehicle-end impedance $Z_{VA}$ is as follows:

$$Z_{VA} = \left(R_L + j\omega L_{f2}\right) // \left(-j\frac{1}{\omega C_{f2}}\right) - j\left(\frac{1}{\omega C_2}\right) \tag{1}$$

The ground end impedance $Z_{GA}$ can be expressed as follows:

$$Z_{GA} = \frac{X_M{}^2 \text{Re}(Z_{VA})}{\text{Re}(Z_{VA})^2 + [\text{Im}(Z_{VA}) + X_2]^2} +$$
$$j\left[X_1 - \frac{X_M{}^2(\text{Im}(Z_{VA}) + X_2)}{\text{Re}(Z_{VA})^2 + [\text{Im}(Z_{VA}) + X_2]^2}\right] \tag{2}$$

where $X_{1/2} = \omega L_{1/2}$, $X_M = \omega M$. According to Formulas (1) and (2), $Z_{VA}$ and $Z_{GA}$ can represent all variable electrical parameters of the system.

$$Z_{SA} = \left(Z_{GA} - j\frac{1}{\omega C_1}\right) // \left(-j\frac{1}{\omega C_{f1}}\right) + j\omega L_{f1} \tag{3}$$

The output power $P_{out}$ of the system can be expressed as follows:

$$P_{out} = I_L{}^2 R_L \approx I_2{}^2 \text{Re}(Z_{VA}) \approx I_1{}^2 \text{Re}(Z_{GA}) \approx I_S{}^2 \text{Re}(Z_{SA}) \tag{4}$$

Furthermore, the transmission efficiency of the system is as follows:

$$\eta = \frac{I_2{}^2 \text{Re}(Z_{VA})}{P_{in}} = \frac{I_1{}^2 \text{Re}(Z_{GA})}{P_{in}} = \frac{I_S{}^2 \text{Re}(Z_{SA})}{P_{in}} \tag{5}$$

In conclusion, $Z_{VA}$, $Z_{GA,}$ and $Z_{SA}$ can characterize the characteristics of the electric vehicle WPT system.

Based on Formulas (4) and (5), it can be seen that there is a corresponding relationship between $P_{out}$ and $\eta$. When the efficiency is constant, the size of Pout depends on the size of the system current and voltage, which is basically a comprehensive consideration of the current-carrying and voltage withstand capacity of the system components. Therefore, this paper focuses on the discussion of transmission efficiency as the final consideration of system interoperability.

According to the particularity of region division in Figure 3, the transmission efficiency of the system can be expressed as follows:

$$\eta = \frac{P_{out}}{P_{in}} = \frac{P_{out}}{\sqrt{P_{out}{}^2 + Q^2}} = \frac{1}{\sqrt{1 + (Q/P_{out})^2}} = \frac{1}{\sqrt{1 + \left(\frac{I_{Z'}{}^2 \text{Im}(Z')}{I_{Z'}{}^2 \text{Re}(Z')}\right)^2}} = \frac{1}{\sqrt{1 + (\text{Im}(Z')/\text{Re}(Z'))^2}} \tag{6}$$

where Z′ is the impedance of a certain port, and the specific parameters are determined according to the actual analysis content. If A = Im (Z′)/Re(Z′) is the detuning coefficient, then the transmission efficiency of the system is obtained as η. The expression is as follows:

$$\eta = \frac{1}{\sqrt{1 + A^2}} \tag{7}$$

The analysis of $A$ shows that $A$ is negatively correlated with $\eta$. When a = 0 and η = 1, the system efficiency is the highest and the interoperability of the system is the best. Therefore, the smaller $A$ is, the easier the system can achieve interoperability. According to the requirements of efficiency in system interoperability in Table 2, the efficiency of vehicles in the right position shall not be less than 85%, and the efficiency shall not be less than 80% in case of offset. It can be inferred that when the vehicle is in the right position, $A$ shall meet the requirements of $-0.62 \le a \le 0.62$, and when the vehicle is offset, $A$ shall meet the requirements of $-0.75 \le a \le 0.75$. Therefore, the curve of $\eta$ about A can be drawn according to Formula (7), and the scope of system interoperability can be defined as shown in Figure 4.

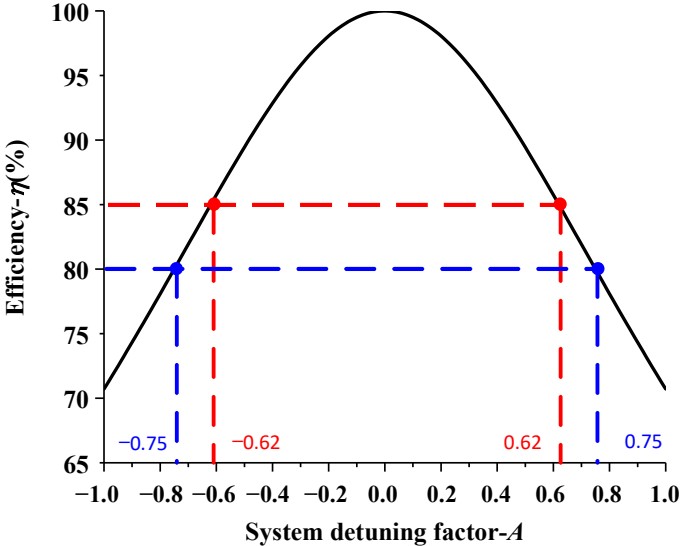

**Figure 4.** Relationship between system detuning factor $A$ and transmission efficiency $\eta$.

For the system output power, $P_{out}$ can also be expressed as:

$$\begin{aligned} P_{out} &= I_S{}^2 \text{Re}(Z_{SA}) = \left(\frac{U_S}{Z_{SA}}\right)^2 \text{Re}(Z_{SA}) \\ &= U_S{}^2 \frac{\text{Re}(Z_{SA})}{\text{Re}(Z_{SA})^2 + \text{Im}(Z_{SA})^2} \end{aligned} \tag{8}$$

Set $\varepsilon$ = Re $(Z_{SA})$/(Re $(Z_{SA})^2$ + Im $(Z_{SA})^2$) as the output power characteristic impedance, the system output power $P_{out} = U_1{}^2 * \varepsilon$, the system output and $\varepsilon$ are positively correlated, when the input voltage is determined to be 200 V. When the $\varepsilon$ is different, the system can achieve the interoperability of different power levels. According to power level requirements, the relationship between system power and $\varepsilon$ can be drawn as shown in Figure 5.

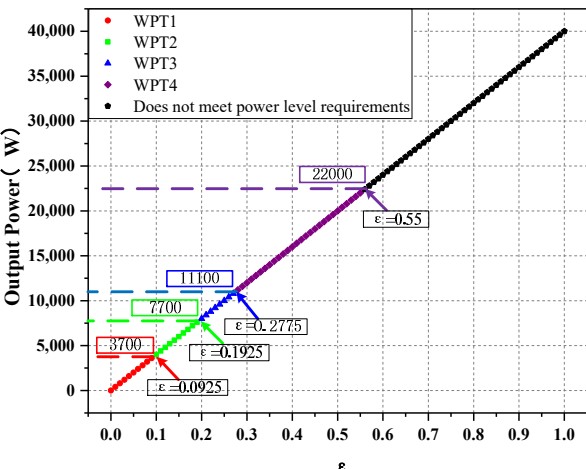

**Figure 5.** Relationship between system power characteristic impedance $\varepsilon$ and transmission power $P_{out}$.

## 3. System Interoperability Evaluation Method

For the actual application scenario of electric vehicle WPT, the ground power supply equipment is usually fixed, and different types of vehicles are parked at the charging position for the WPT process; At the same time, combined with the two parts of the evaluation of electric vehicle WPT system, the evaluation process first evaluates the compensation network at the vehicle end, and then evaluates the coupling coil. When the parameters of the on-vehicle compensation network are different or the coil is shifted, the original resonant system will have reactive power components, which will affect the interoperability of the system. In this process, reactive power becomes the main part of the system interoperability. The KVL equation of the second and third mesh loops is as follows:

$$\begin{cases} -\frac{1}{\omega C_{f1}} I_S = \left(\omega L_1 - \frac{1}{\omega C_1} - \frac{1}{\omega C_{f1}}\right) I_1 - \omega M I_2 \\ \omega M I_1 = \left(\omega L_2 - \frac{1}{\omega C_2} - \frac{1}{\omega C_{f2}}\right) I_1 + \frac{1}{\omega C_{f2}} I_L \end{cases} \tag{9}$$

It can be seen from Equation (9) that if the front circuit of the system meets the resonance condition $\omega^2 (L_1 - L_{f1}) C_1 = 1$, $\omega^2 (L_2 - L_{f2}) C_2 = 1$, then Formula (9) can be converted into:

$$\begin{cases} \dfrac{1}{\omega C_{f1}} I_S = \omega M I_2 \\ \omega M I_1 = \dfrac{1}{\omega C_{f2}} I_L \end{cases} \tag{10}$$

Formula (10) indicates that the input energy of the system can be transferred to the secondary circuit without loss. However, in actual operation, the difference of on-board compensation parameters and vehicle offset will lead to the front and rear circuit detuning, which will affect the interoperability of the system. To represent the change of reactive power parameters in system regions 1 and 2, remember $\Delta X_1 = \omega (L_1 - L_{f1}) - 1/(\omega C_1)$ is the former detuning reactance, $\Delta X_2 = \text{Im}(Z_{VA}) \pm X_2 = \omega (L_2 - L_{f2}) - 1/(\omega C_2)$ is the second step detuning reactance and according to the circuit relationship and the previous analysis, the detuning reactance is $\Delta X = \Delta X_1 + (\omega^3 M C_{f22} R_L)^2 \Delta X_2$. The generation of detuning reactance will lead to the reduction of system performance and affect the system interoperability.

### 3.1. Interoperability Evaluation of System Efficiency

For wireless charging systems, $Z_{VA}$, $Z_{GA}$, and $Z_{SA}$ can be used to characterize system efficiency and interoperability. However, it will have certain limitations for $Z_{VA}$ and $Z_{GA}$: the use of $Z_{VA}$ characterization can only reflect the interoperability of the on-vehicle compensation network. The impact of interoperability due to charging position error cannot be reflected; using $Z_{GA}$ to characterize the interoperability impact of the on-board compensation network, the transceiver coil can be reflected. However, it cannot be characterized

when there is a problem with the ground-side compensation network. In addition the method based on $Z_{VA}$ and $Z_{GA}$ to characterize interoperability cannot be applied at the same time. This is because when the detuning coefficient corresponding to $Z_{VA}$ meets the system efficiency requirements, the detuning coefficient corresponding to $Z_{GA}$ does not necessarily meet the system efficiency requirements. Detuning coefficients do not properly characterize system efficiency interoperability. Therefore, in order to be unified with the evaluation parameters of system efficiency interoperability, it was finally decided to use $Z_{SA}$ parameters to evaluate the system efficiency interoperability.

The total output power of the system is expressed as:

$$
\begin{aligned}
P_{in} &= \sqrt{P^2 + Q^2} \\
&= I_S^2 \sqrt{\mathrm{Re}(Z_{SA})^2 + \mathrm{Im}(Z_{SA})^2}
\end{aligned}
\tag{11}
$$

The overall efficiency of the system is expressed as:

$$
\eta = \frac{1}{\sqrt{1 + \left(\frac{\mathrm{Im}(Z_{SA})}{\mathrm{Re}(Z_{SA})}\right)^2}}
\tag{12}
$$

The system detuning coefficient $A$ is defined as:

$$
A = \frac{\mathrm{Im}(Z_{SA})}{\mathrm{Re}(Z_{SA})}
\tag{13}
$$

Formula (12) is converted into:

$$
\eta = \frac{1}{\sqrt{1 + A^2}}
\tag{14}
$$

According to Formula (14), $|A|$ is negatively correlated with $\eta$, and its functional relationship is shown in Figure 4. The curve $A$ is drawn on the Re $(Z_{SA})$ – Im $(Z_{SA})$ impedance plane as shown in Figure 6. $A$ should satisfy the condition of $-0.62 \leq A \leq 0.62$ when the vehicle is facing the position, and $-0.75 \leq A \leq 0.75$ when the vehicle is offset. It should be noted that when Im $(Z_{SA}) > 0$, this means that area 2 is inductive, and the corresponding Re $(Z_{SA})$ axis should be located on the positive semi-axis of Im $(Z_{SA})$; when Im $(Z_{SA}) < 0$, this means that area 1 is capacitive, the corresponding Re $(Z_{SA})$ axis should be located on the negative semi-axis of Im $(Z_{SA})$.

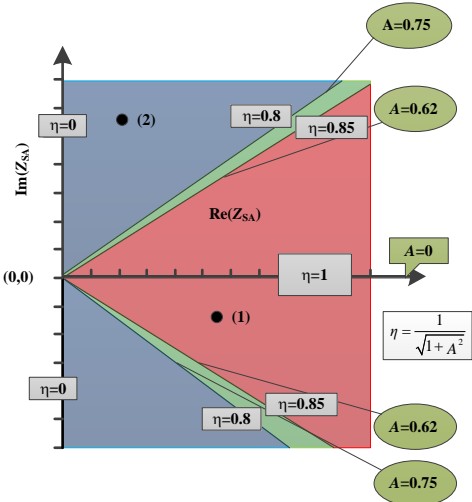

**Figure 6.** Schematic diagram of *A-η* on $Z_{SA}$ impedance plane.

### 3.2. Interoperability Evaluation of System Output Power

The input and out power of the system can be calculated from the topology in Figure 3.

$$P_{in} = U_S I_S = I_S{}^2 Z_{SA} \tag{15}$$

$$
\begin{aligned}
P_{out} &= I_S{}^2 Re(Z_{SA}) = \left(\frac{US}{Z_{SA}}\right)^2 Re(Z_{SA}) \\
&= US^2 \frac{Re(Z_{SA})}{Re(Z_{SA})^2 + Im(Z_{SA})^2}
\end{aligned} \tag{16}
$$

Set Re $(Z_{SA})/($Re $(Z_{SA})^2 +$ Im $(Z_{SA})^2)$ to be the power characteristic impedance $\varepsilon$. When the system input voltage is determined, the system output power is positively correlated with $\varepsilon$. The relationship is shown in Figure 5. When the voltage is 200 V, when $\varepsilon \leq 0.0925$, the system meets the requirements of WPT1. When $0.1925 \geq \varepsilon > 0.0925$, the system meets the requirements of WPT2. When $0.2775 \geq \varepsilon > 0.1925$, the system meets the requirements of WPT3. When $0.55 \geq \varepsilon > 0.2775$, the system meets the requirements of WPT3 and meets the WPT4 requirements. Other $\varepsilon$ values do not meet the system power level requirements.

To sum up, when evaluating whether the WPT system of an electric vehicle has interoperability, the compensation network of the electric vehicle should be evaluated first, which is a product of self-inspection for the manufacturer, and then the interoperability of the coupling coil should be evaluated. When the condition of the system detuning coefficient $A$ is satisfied, it is proved that the electric vehicle WPT system has transmission efficiency interoperability. When the system meets the condition of power characteristic impedance $\varepsilon$, it is proved that the electric vehicle WPT system has output power interoperability. The specific evaluation process of interoperability is shown in Figures 7 and 8.

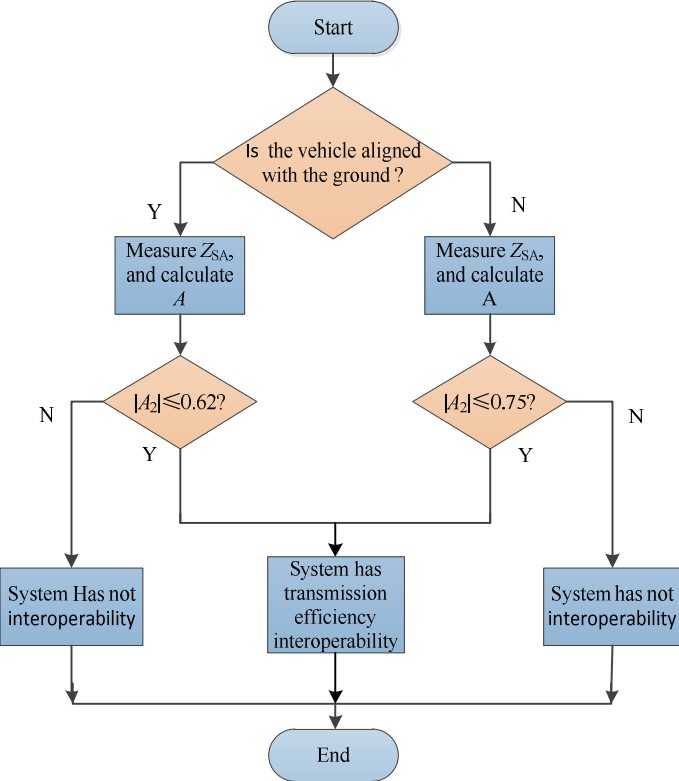

**Figure 7.** Flow chart of efficiency interoperability evaluation of WPT system for electric vehicles.

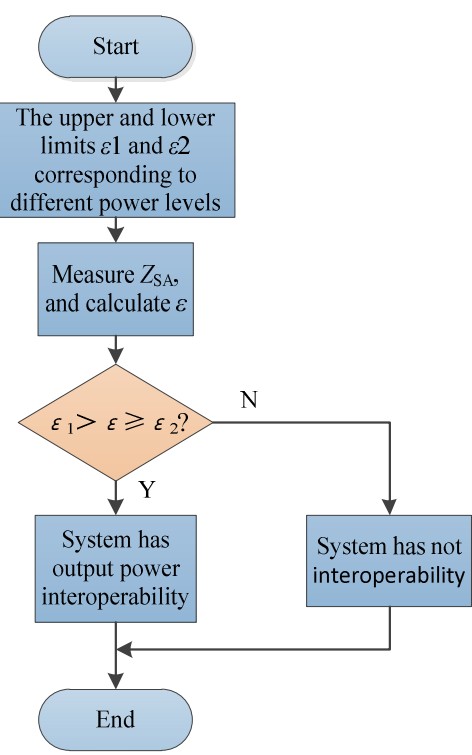

**Figure 8.** Flow chart of output power interoperability evaluation of electric vehicle WPT system.

## 4. System Interoperability Simulation

In order to verify the correctness of the above interoperability evaluation methods, the simulation modeling of 7.7 kW (WPT2) high power is carried out according to the analysis of the previous evaluation method, and the simulation verification of the previous interoperability method is carried out in turn. First, the coil parameters of the ground terminal and the vehicle terminal are determined, and then according to the resonance conditions of the system $\omega^2 L_{f1/2} C_{f1/2} = 1$, $\omega^2 (L_{1/2} - \underline{L}_{f1/2} C_{f1/2}) = 1$ the values of each parameter in the system in turn are determined. To simulate system parameter differences, if the selected range is too large, it will lead to serious system detuning and large performance deviation. The calculation results and range selection of system simulation parameters are shown in Table 3. Scanning for $C_2$, $C_{f2}$, $L_{f2}$, the scanning conditions are $C_2$: range (55, 1, 65); $C_{f2}$: (220, 3, 240); $L_{f2}$: (1, 3, 20), the scanning results are shown in Figures 9 and 10.

**Table 3.** System simulation parameters.

| Parameter | Standard Value | Value Range | Company |
|---|---|---|---|
| $U_S$ | 200 | — | V |
| f | 85,500 | — | Hz |
| k | 0.25 | (0.1~0.3) | — |
| $L_{f1}$ | 18 | — | μH |
| $C_{f1}$ | 192.5 | — | nF |
| $C_1$ | 157.5 | — | nF |
| $L_1$ | 40 | (35~45) | μH |
| $L_2$ | 70 | (65~75) | μH |
| $C_2$ | 63 | (61~65) | nF |
| $C_{f2}$ | 231 | (220~240) | nF |
| $L_{f2}$ | 15 | (10~20) | μH |
| $R_L$ | 12 | — | Ω |

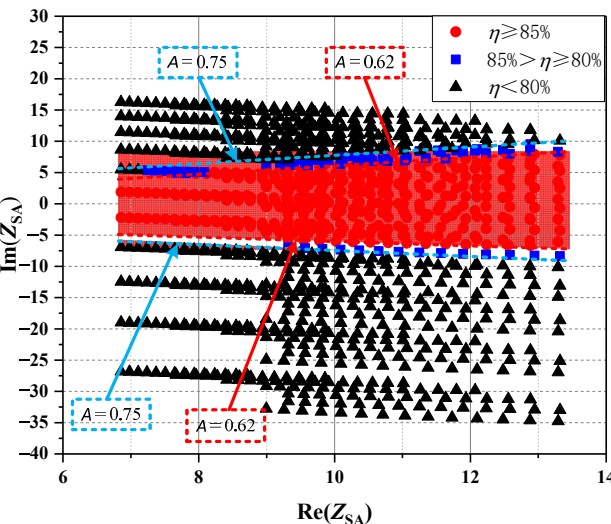

**Figure 9.** System transmission efficiency interoperability simulation.

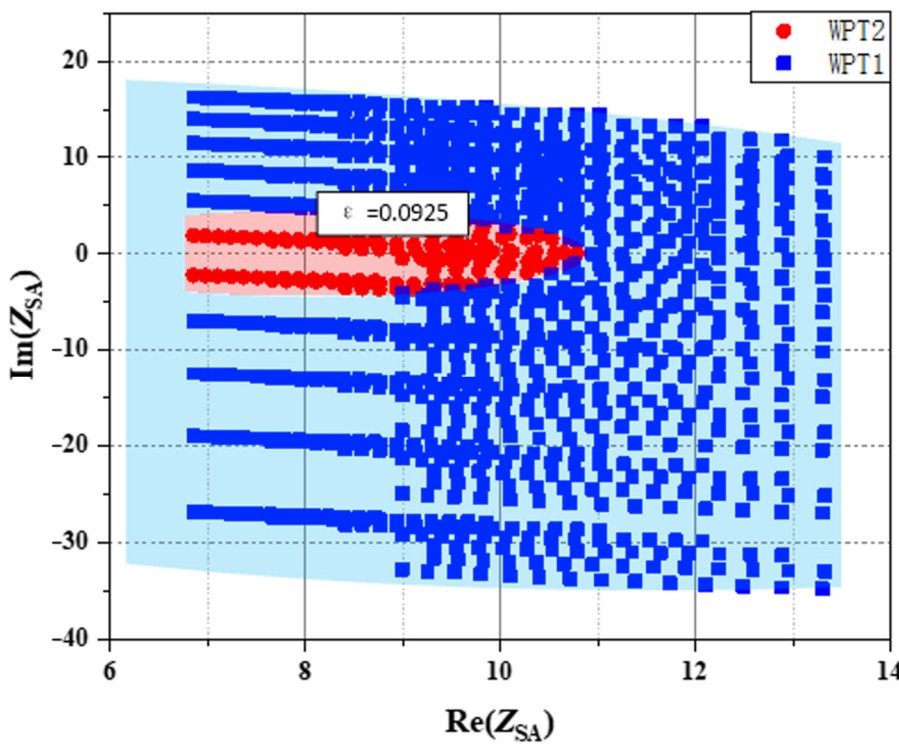

**Figure 10.** System output power interoperability simulation.

### 4.1. System Transmission Efficiency Interoperability Simulation

It can be seen from the simulation results in Figure 9 that when the various parameters of the system vary near the resonance point, the efficiency at the efficiency point is when the horizontal axis Im $(Z_{SA})$ = 0 is the highest, because the $Z_{SA}$ at this time is purely resistive and the system input power has no reactive component. With the increase of the offset angle, the $Z_{SA}$ gradually becomes capacitive or inductive, the reactive component of the input power becomes larger and larger, and the system transmission efficiency gradually decreases. $A$ = 0.62/0.75 is the demarcation point of $\eta$ = 85%/80%, which are marked with a red line/blue line respectively in the figure.

*4.2. System Output Power Interoperability Simulation*

It can be seen from the simulation results in Figure 10 that when the input voltage is 200 V, the system parameters can only satisfy the results of two power levels when the system parameters change near the resonance point, and the output power is the largest when the horizontal axis Im ($Z_{SA}$) = 0. At this time, $Z_{SA}$ is purely resistive, and there is no reactive power in the output power of the system; with the increase of $\varepsilon$, the value of $Z_{SA}$ gradually increases, so that the output power of the system gradually decreases. It is divided into WPT1 and WPT2 with $\varepsilon$ = 0.0925 as the dividing line of the two power levels.

## 5. WPT System Interoperability Experiment

In order to verify the correctness of the theoretical analysis, an experimental platform for WPT interoperability of electric vehicles was built. Due to the limitation of experimental conditions, it is impossible to build a high-power experimental platform. According to the existing conditions, a WPT interoperability experimental platform for electric vehicles with a maximum transmission efficiency of 1 kW was built for power-on experiments. The experimental equipment mainly includes the following: power supply, test bench, ground end compensation network, vehicle carrier end compensation network, transmitting coil, receiving coil, LCR measuring instrument, load, etc., as shown in Figure 11. The experimental parameters are consistent with the parameters in Table 3 except that the input voltage is 100 V.

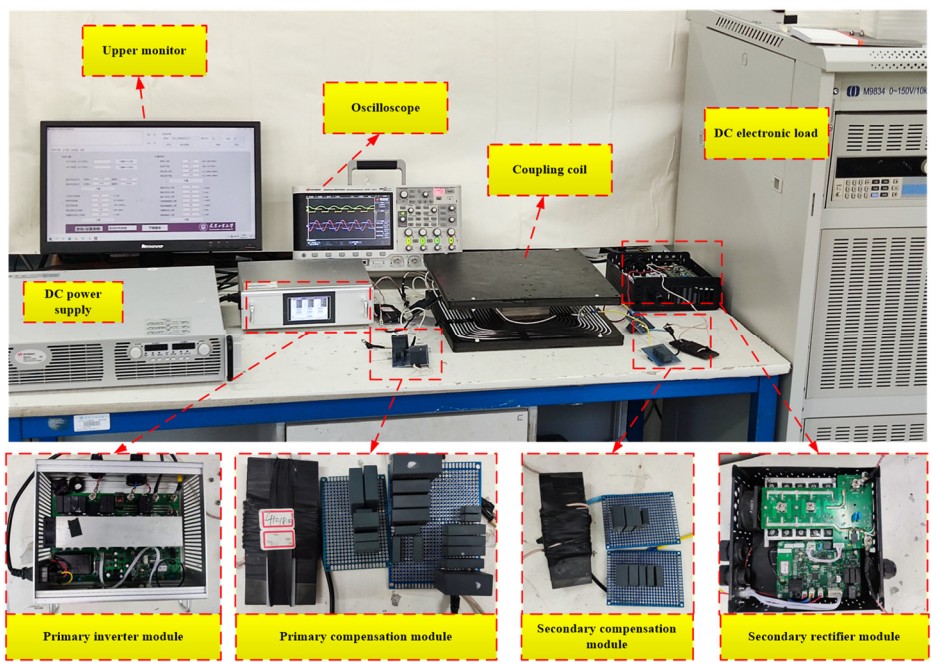

**Figure 11.** Electric vehicle WPT interoperability experimental system.

*Interoperability Experiment Parameters and Results*

The interoperability of the electric vehicle wireless charging system is mainly divided into the interoperability of the compensation network and the interoperability between the coupling coils. The parameters are selected for this. Select $C_2$, $C_{f2}$, $L_{f2}$ as variables for the compensation network, and select $L_1$, $L_2$, k as variables for the coupling coil. In order to keep the experimental environment safe and the experimental results reasonable, all the data are selected near the resonance point, and the simulation transmission efficiency results of different groups are distributed at various intervals, while the errors of the experiment and simulation are compared under different conditions. The experimental data are shown in Tables 4 and 5.

**Table 4.** Multi group compensation network parameters.

| Group | $C_2$ (nF) | $C_{f2}$ (nF) | $L_{f2}$ (μH) |
|:-----:|:----------:|:-------------:|:-------------:|
| 1 | 63 | 225 | 20 |
| 2 | 63 | 225 | 25 |
| 3 | 70 | 240 | 10 |
| 4 | 55 | 230 | 12 |

**Table 5.** Multi group coupling coil parameters.

| Group | $L_1$ (μH) | $L_2$ (μH) | $k$ |
|:-----:|:----------:|:----------:|:-----:|
| 1 | 42 | 68 | 0.198 |
| 2 | 43.5 | 68 | 0.15 |
| 3 | 43.5 | 68 | 0.17 |
| 4 | 44 | 70 | 0.21 |

From the experimental results in Figure 12, it can be seen that when the parameters of the experiment and simulation are consistent, the maximum error between the experimental data and the simulation data in terms of transmission efficiency is 5.62%, while the average error is 4.94%. In terms of output power, the maximum error between the experimental data and the simulation data is 7.44%, and the average error is 6.74%. Analyzing the experimental results of Figure 13, it can be seen that in terms of transmission efficiency, the experimental data results are better, the maximum error is 8.57%, and the average error is 6.63%. In terms of output power, the experimental data and simulation data, the maximum error is 3.37%, and the average error is 1.29%. In terms of transmission efficiency interoperability, when the experimental and simulated detuning coefficients *A* are consistent and the interoperability evaluation results of other groups except group 4 in Figure 12a and group 3 in Figure 13a, the wireless charging efficiency interoperability evaluation can be consistently performed correctly. Due to the limitation of the experimental conditions, the internal resistance of the circuit and the coil is larger than that in the actual application, so the two groups with poor results in the experiment are more efficient in practical applications. The result is larger, which verifies the correctness of the detuning coefficient *A*; in terms of output power, because the maximum output power of the experimental platform built is 1 kW, the correctness of the output power interoperability evaluation cannot be judged according to the power level requirements, but the experimental error of the results is small. The average errors of the two output efficiency interoperability experiments are 6.74% and 1.29%, respectively. The experiment is basically consistent with the theory, and the correctness of the power characteristic impedance *ε* can be verified.

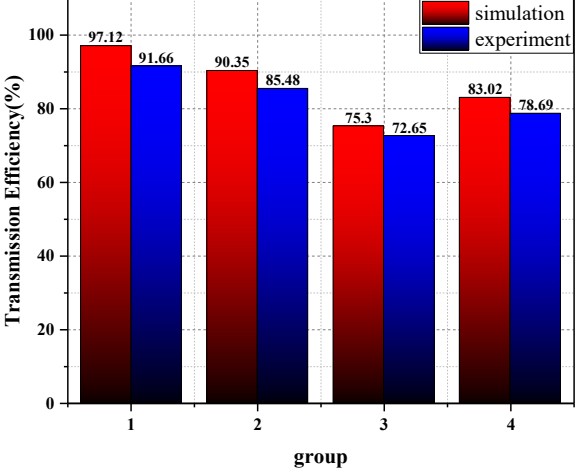

(**a**) Comparison of transmission efficiency

**Figure 12.** *Cont.*

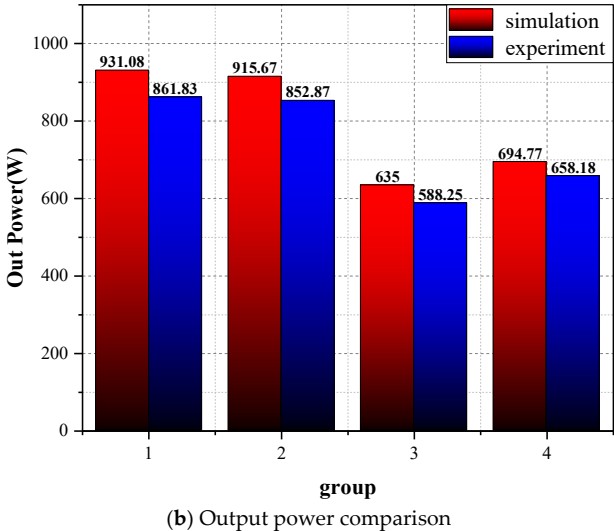

(**b**) Output power comparison

**Figure 12.** Compensation network interoperability experimental results.

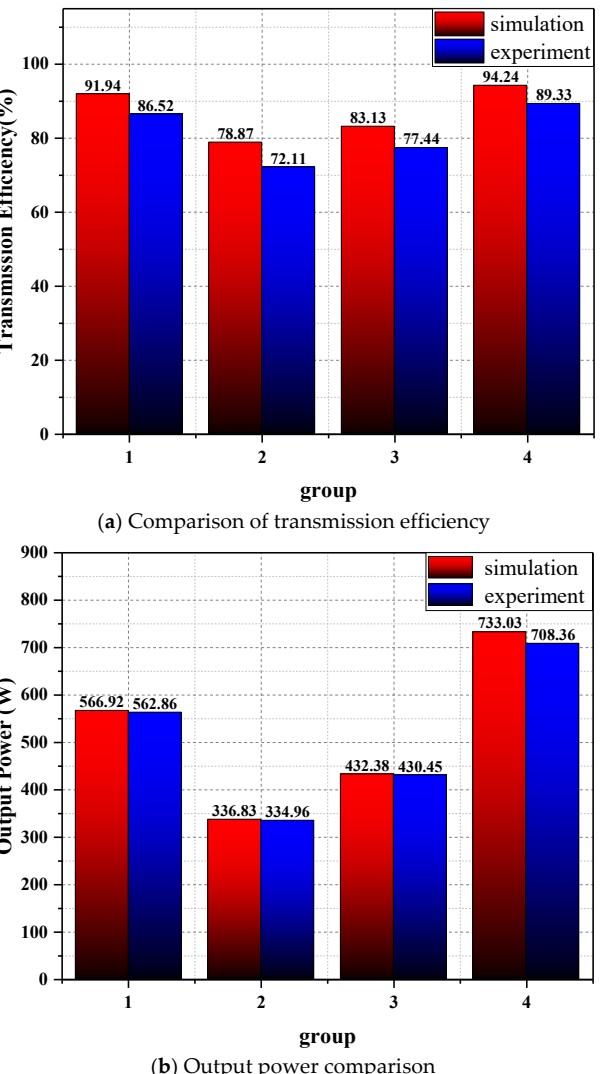

(**a**) Comparison of transmission efficiency

(**b**) Output power comparison

**Figure 13.** Experimental results of coupled coil interoperability.

## 6. Conclusions

In this paper, a dual-parameter characterization evaluation method is proposed for the interoperability of dual-LCC compensation networks in electric vehicle WPT systems. First, the influence of system parameters on the system interoperability is analyzed by the impedance method, and the parameter $A$ is proposed from the defined $Z_{SA}$ two-port impedance. Then the parameter $A$ can be used to evaluate the interoperability of the transmission efficiency of the system; the parameter power is proposed. The characteristic impedance $\varepsilon$ can be used to evaluate the interoperability of the system output efficiency through the parameter $\varepsilon$. The electric vehicle WPT system interoperability experimental system platform is built. By changing the parameters in the compensation network, the relationship between the system transmission efficiency and the parameter $A$ is analyzed, and the deduced theory is verified by combining with simulation. It was proved that this interoperability evaluation method was correct. Based on the research of predecessors, this paper proposes a new interoperability evaluation method, which provides a new direction and a new idea the interoperability evaluation research. It provides a reference for the promotion of research work on the interoperability of the wireless charging system for electric vehicles.

**Author Contributions:** Writing—original draft preparation L.S. and J.L.; writing—review and editing, L.S. and Z.C.; data curation, Z.C. All authors have read and agreed to the published version of the manuscript.

**Funding:** This work is supported by The National Natural Science Foundation of China (No. 51807138), The National Natural Science Foundation of China (No. 52077153).

**Institutional Review Board Statement:** Not applicable.

**Data Availability Statement:** Not applicable.

**Conflicts of Interest:** The authors declare no conflict of interest.

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
