# Peer review of "Research on Evaluation Method of Electric Vehicle Wireless Charging Interoperability Based on Two Parameter Representation"

_processes, doi:10.3390/pr10081591_

Round 1
Reviewer 1 Report
The proposal is interesting, but there are specific observations such as:
1. The Pout expression indicated in (4), is only correct if the efficiency is equal to 100%, in this type of systems this is not satisfied, so it should be indicated what will be the impact on the proposal if the efficiency is less than 100%.
2. Equation (6). It is necessary to indicate the development of the expressions Q/Pot=Im(Z')/Re(Z').
3. It is not clear how ∆X1, ∆X2 and ∆X2 and ∆X are obtained and what is their importance in the proposal.
It is not clear how the values indicated in Fig 5 are obtained. The authors indicate that "the input voltage is determined to be 200V", but observing Fig 5, the values of ε do not correspond with what is indicated in the Output Power axis.
5. Figure 8. ε1 and ε2, what do they represent?
6. Figure 9. It should be indicated in the writing which parameter was varied with respect to the nominal value, to achieve what is indicated in the Re(ZA) axis. The way the authors present it is confusing.
7. Fig. 10, It should be indicated in the wording which parameter was varied with respect to the nominal value, in order to achieve what is indicated on the Re(ZA) axis. As presented by the authors it is confusing.
8. In the WPT system interoperability experiment section, the way in which the results are presented is confusing.
For the case study Multi group compensation network parameters, it is varying to ZVA, but the authors do not justify the values indicated in Table 4. This part should be clearly indicated with minimal writing, so that the data presented in Fig. 12 can be understood.
For the case study Multi group coupling coil parameters, the same is requested.
Revise the format of the whole article, since there is no separation between the final point and the following sentence, change "company" for units.
Author Response
- Pout in the formula is also true when the efficiency is not 100%. This formula expresses the active power consumed by different characteristic impedances, because in the case of ignoring the internal resistance of the coil and the internal resistance of the line, the system has only one resistance RL. , so the calculation of active power from different characteristic impedances is actually consumed by RL. I also verified this conclusion through simulation.
- For formula (6), I am not sure if you feel that the process of derivation from pout/Pin to the final result is not very clear. If so, I can explain it here. The formula mainly analyzes the input and output power. The relationship between active power and reactive power ratio, but this formula can only be established if the compensation structure in the previous item is kept in resonance, which has limitations. This is also the reason for extracting the detuning coefficient in ZSA. In addition, the derivation in the formula is not clear. I have edited it, I don't know if it meets your requirements.
- X2 is the inductance value of the on-vehicle coil, which appears for the first time in formula (2), ΔX2 and ΔX are the imaginary parts of the characteristic impedance of the on-vehicle part and the ground part.ω2(L2- Lf2) C2 = 1 is derived, which reflects the resonance of different parts of the system, which is helpful for the calculation of the detuning coefficient A.
- According to the formula Pout= U1²*ε, because the output power is related to the voltage and ε, ε is only instructive when the voltage is determined. Figure 5 uses ε as the independent variable and voltage as the dependent variable according to the previous formula. It represents the area that meets the requirements of different power levels, and has been modified for the situation that the picture is unclear.
- ε1 and ε2 represent the upper and lower limits of ε corresponding to the power level you require, as shown in Figure 5. For example, when the input voltage is 200v, I require the power level to be WPT2, then my ε1 is 0.2775, and ε2 is 0.1925 .
- The picture in Figure 9 is obtained by changing the parameters C2, Cf2, Lf2 of the on-board terminal on the basis of resonance, C2 (55, 1, 65); Cf2 (220, 3, 240); Lf2 (1, 3, 20).
- The parameter changes in Figure 10 are the same as the above figure. Each parameter change corresponds to an output power, and the corresponding Re(ZSA) and Im(ZSA) will be obtained. Put the simulation results on the ZSA impedance plane to see ε Relationship with ZSA
- The values in Table 4 are selected near the resonance point. If the selection range is too large, the detuning will be serious and the performance shift will be too large, and the research will be meaningless; Another reason is that the simulation transmission efficiency results are distributed in different intervals,such like <80%;80%-85%;>85%.It's shown in the simulation data in Figure 12(a), the same selection principle is also used for the data in Table 5.
In response to your comments, I have added the parts that need to be revised and explained in the original text and marked in red. Thank you for your review. It is also very helpful for my future writing.

Reviewer 2 Report
1-Authors must show how their paper differs and improves upon previous formulations.
2-The authors should clearly describe the contributions.
3-Please add one general flowchart of the proposed method to manuscript.
4. Avoid using long sentences.
5.In the conclusions, in addition to summarizing the actions taken and results, please do explain their significance.
Author Response
1 Compared with the previous formulations, the innovation in this paper is that a new characteristic impedance ZSA is specified on the basis of the characteristic impedance method, and based on ZSA, the detuning coefficient A and the power characteristic impedance ε are proposed for the transmission of the system. The interoperability evaluation of efficiency and power provides a new evaluation method for electric vehicle interoperability research.
2 has been added to the original text
3 The flowchart of the proposed interoperability evaluation method is shown in Figure 7 and Figure 8
4 has been modified from the original
5 has been added to the original text
